# Weight Change and Cardiometabolic Outcomes in Postpartum Women with History of Gestational Diabetes

**DOI:** 10.3390/nu11040922

**Published:** 2019-04-24

**Authors:** Siew Lim, Vincent L. Versace, Sharleen O’Reilly, Edward Janus, James Dunbar

**Affiliations:** 1Monash Centre for Health Research and Implementation, Monash University, 3168 Clayton, Australia; 2Deakin Rural Health, School of Medicine, Deakin University, 3217 Geelong, Australia; vincent.versace@deakin.edu.au (V.L.V.); james.dunbar@deakin.edu.au (J.D.); 3Institute of Food and Health, School of Agriculture and Food Science, University College Dublin, Belfield, Dublin 4, Ireland; sharleen.oreilly@ucd.ie; 4General Internal Medicine Unit, Western Health and Department of Medicine, Melbourne Medical School—Western Precinct, University of Melbourne, 3010 Melbourne, Australia; edwarddj@unimelb.edu.au

**Keywords:** gestational diabetes, diabetes prevention program, weight loss, weight gain, postpartum

## Abstract

Weight gain after childbirth is a significant risk factor for type 2 diabetes (T2DM) development after gestational diabetes mellitus (GDM). The level of weight loss achieved in diabetes prevention programs for women after GDM is often low but its effects on the cardiometabolic risk are not known. In a secondary analysis of a diabetes prevention program in postpartum women with history of gestational diabetes, we evaluated the effect of weight change on the cardiometabolic outcomes at 1-year follow-up. Of the 284 women randomized to the intervention arm, 206 with the final outcome measurements were included in the analyses. Participants were categorized into weight loss (>2 kg, *n* = 74), weight stable (±2 kg, *n* = 74) or weight gain (>2 kg, *n* = 58) groups. The weight loss group had significantly greater decrease in glycated hemoglobin (HbA1c) than the weight gain group (−0.1 + 0.4% vs. 0 + 0.4%, *p* = 0.049). The weight loss group had significantly greater decrease in total cholesterol and low-density lipoprotein cholesterol cholesterol than the other two groups (*p* < 0.05). The weight gain group had significantly greater increase in triglyceride and triglyceride:high-density lipoprotein cholesterol ratio compare with the other groups (*p* < 0.01). Overall, a small amount of weight loss and prevention of further weight gain was beneficial to the cardiometabolic outcomes of postpartum women after GDM.

## 1. Introduction

Gestational diabetes mellitus (GDM) affects one in five pregnancies according to the International Association of the Diabetes and Pregnancy Study Groups (IADPSG) criteria [1]. This could rise to above 40% in certain ethnic groups such as Middle Eastern populations [2]. Women with previous GDM are seven times more likely to develop type 2 diabetes (T2DM) compared with women with normoglycemic pregnancies [3]. Weight gain after childbirth is a significant risk factor for T2DM development after GDM; the incidence of T2DM increases incrementally with weight gain [4,5]. Due to the risks associated with obesity in the development of T2DM, weight loss is one of the intervention goals in diabetes prevention programs for the general population and for women after GDM [6,7,8].

The development of T2DM in high-risk groups can be prevented or delayed with lifestyle interventions. Numerous real-world diabetes prevention programs in the general population have demonstrated a significant reduction in the incidence of diabetes through diet and physical activity modifications [9,10]. Both the Finnish Diabetes Prevention Study and United States Diabetes Prevention Program (DPP) reported a reduction diabetes incidence of up to 58% with approximately 5 kg weight loss in their lifestyle intervention arms [6,11]. Younger populations such as those targeting women after GDM report much smaller weight loss, averaging between 1 to 3 kg, with correspondingly smaller effects on the reduction of diabetes incidence (relative risk 0.75, 95% confidence interval (CI) 0.55–1.03) [12]. Greater weight regain among younger participants (45 years and below) has also been reported in a 10-year follow up study of the US DPP study [13]. The challenges with weight loss and weight loss maintenance for younger women in diabetes prevention programs are consistent with longitudinal studies in the general population, which showed that women between 18 to 40 years are at higher risk of weight gain compared with older women [14,15]. The secular trend of weight gain in this age group may have attenuated the weight loss effect of diabetes prevention programs in younger women with histories of GDM. In addition, a sub group analysis of the US DPP comparing women who previously had GDM on average 12 years before with women who had not had GDM, the women who previously had GDM were much less likely to respond to lifestyle modification [16].

The reasons for weight gain in women of childbearing age are complex. Being married, having children and starting work are risk factors for weight gain in younger women [17]. These changes often result in increased sedentary behaviours, decreased physical activity and a worsening of dietary habits [17,18]. Among postpartum women, a lack of time, tiredness, financial constraints and changes in priorities after childbirth further contribute to the barriers to a healthy lifestyle [19,20]. Interventions may not be able to address many of these barriers. While higher intervention intensity may increase weight loss, the presence of contextual barriers to lifestyle modification in younger women has meant that higher intensity interventions are met by low participation or retention rates [21,22]. Thus, interventions with lower intensity and a smaller weight loss goal may be more feasible for this group. However, a paucity of data on the metabolic effect of lower-level weight loss or weight gain prevention in women after GDM remains.

Owing in part to the lower weight loss achieved in diabetes prevention programs for women after GDM, these programs also tend to produce smaller, inconsistent and often insignificant improvements in glycemic markers such as glycated hemoglobin (HbA1C) or fasting plasma glucose [12,23]. As insulin resistance and hyperinsulinemia precedes hyperglycemia in the natural history of T2DM development, measuring changes in markers of insulin resistance may more appropriate and relevant than changes in glycemic markers when measuring progress in diabetes prevention in younger cohorts [24]. Recently, triglyceride-to-high-density lipoprotein cholesterol ratio (triglyceride:HDL) has been suggested to be marker of insulin resistance in some population groups [25,26]. Changes in triglyceride:HDL was not previously investigated in diabetes prevention programs.

Considering the effect of postpartum weight gain on the development of T2DM [4,5], prevention of weight gain may be efficacious in preventing diabetes in women after GDM. There is little direct evidence available to confirm the benefit of weight gain prevention on the cardiometabolic risk of women after GDM. Consistently lower-level weight loss achieved in diabetes prevention programs in this group suggests that greater weight loss may not be feasible at a large scale in this population. This necessitates a greater understanding on the metabolic benefits of small weight changes in this group within the context of diabetes prevention. A stratified approach to investigate the effect of lower-level weight changes on cardiometabolic effects has not been previously investigated in women after GDM. Therefore, this study aims to compare the cardiometabolic effects of weight loss, weight maintenance and weight gain following a diabetes prevention program in women after GDM.

## 2. Materials and Methods

### 2.1. Trial Design

This is a secondary analysis of a 12-month multicenter, open, randomized controlled trial (RCT). The study protocol and results have been previously published [8,27]. Mothers After Gestational Diabetes in Australia (MAGDA) was developed to reduce the risk of diabetes through dietary and physical activity modifications for women with recent histories of gestational diabetes. Its development has a direct lineage from the Finnish Diabetes Prevention Study, the Greater Green Triangle Diabetes Prevention Program and the Melbourne Diabetes Prevention Study [7,28,29]. The intervention goals of MAGDA adhered to those of the Finnish DPS, i.e., ≥5% weight loss, ≤30% total energy from total fat intake, ≤10% total energy from saturated fat, ≥15 g dietary fibre per 1000 kcals and ≥30 min moderate to vigorous physical activity per day [30]. MAGDA was set across 12 study partners or recruitment sites in the Australian states of Victoria and South Australia, including three universities, five hospitals, two state health departments and two not-for-profit organizations. Participants were recruited through the National Gestational Diabetes Register within the National Diabetes Services Scheme (NDSS), from participating hospitals and through referrals from private healthcare providers. Informed consent was obtained from all participants. All women were followed up at 12 months after the baseline measurements. The intervention group were additionally followed up at 3 months after baseline measurements. The study was approved by the ethics committees at each participating sites (e.g., South Australia Health human research ethics committee. This trial is registered with the Australian New Zealand Clinical Trials Registry ACTRN12610000338066.

### 2.2. Participants

Participants were women 18 years and older with GDM diagnosed using the Australasian Diabetes in Pregnancy Society (ADIPS) criteria in their most recent pregnancy [31]. Exclusion criteria include pre-existing diabetes, cancer, severe mental illness, substance abuse, myocardial infarction in the preceding three months, difficulty with English, involvement in another post-natal intervention trial, pregnancy at any point during the 12 months of study.

### 2.3. Intervention

The intervention consisted of an individual face-to-face session and five group face-to-face sessions at two-week intervals, followed by two telephone calls at 3 and 6 months after the final group session to reinforce behavior change (total number of sessions was eight sessions). The initial individual session was conducted at the participants’ home while the group sessions were conducted at community facilities. The sessions were delivered by healthcare professionals who received training on the program content and practical components of the intervention. In all sessions, facilitators present on a specific topic relating to the intervention goals, followed by group activities that reinforced the skills required to meet the goal. Participants were also guided to set personal goals and review goal progress at each session. The content of the activities were tailored to meet the needs of postpartum mothers and their families, such as preparing reduced fat meal for the whole family.

The MAGDA core curriculum was as previously published [8,27]. The initial individual session covered diabetes and diabetes risk factors, risk perception and an introduction to the five intervention goals. The first group session was on limiting saturated fat intake. The second session was on reducing total fat intake and managing postpartum weight. The third session was on strategies to increase fibre intake and skills on food shopping. The fourth session covered meal planning, negotiating stressful situations around food choice with family members, mindful eating and sleep. The final session covered postnatal depression, stress management and relapse prevention. Each group session lasted approximately 120 min.

### 2.4. Control

The control group received usual care within their local setting. They were not provided with any additional advice or support during the study.

### 2.5. Measurements

The protocol for measurements was described in detail in previous publication [27]. In brief, participants completed a survey that include demographic questions at baseline. Anthropometric and blood pressure measurements were completed by trained study nurses to ensure that measurements meet research standards. Blood samples were collected by study nurse or phlebotomists and analyzed by private pathology centers including Melbourne Pathology (Victoria) or Clinpath Laboratories (South Australia). All clinical measures including height, weight, waist circumference and blood pressure were measured using according to the recommendations of the European Health Risk Monitoring protocol [32]. Fasting venous blood samples were collected after an overnight fast from 10 pm the night before. Blood samples were analyzed for triglycerides, total cholesterol, low-density lipoprotein (LDL) cholesterol and high-density(HDL) cholesterol, HbA1c, and fasting glucose and 2-h glucose tolerance as previously described [8,27].

### 2.6. Randomization

Block randomization was implemented using a computer generated sequence with a separate randomization list for each venue. Participants and field researchers are aware of the allocation once randomized.

### 2.7. Statistical Analyses

Statistical analysis was performed using IBM SPSS Statistics for Windows, Version 24.0, (Armonk, NY, USA). Baseline participant characteristics were described as frequency and percentage, and were stratified by three groups: (1) weight loss >2 kg, (2) weight stable ± 2 kg, and (3) weight gain >2 kg. Differences in the proportions between groups were analyzed using the chi-square test. Cardiometabolic characteristics were reported according to these stratified groups and described as mean and standard deviation. Changes in these characteristics over time within a group (baseline and 12 months) were analyzed using paired sample t-tests. The differences in changes between the three groups were analyzed using a one-way analysis of variance (ANOVA) with least significant difference as the post-hoc test. The differences in changes between the three groups were analyzed using one-way analysis of covariance (ANCOVA) with baseline weight as a covariate. All available data from intervention participants in the MAGDA trial were used where the participant had both a baseline and 12 month follow up measure. All statistical tests were conducted at the 5% significance level with no adjustments for multiple comparisons. Significance was reported where *p*-value was ≤0.05.

## 3. Results

The Consolidated Standards of Reporting Trials (CONSORT) diagram describing the flow of participants is as shown in Figure 1. From a total of 573 participants randomized, 434 completed the study at 12-months. The main results of the RCT were previously published [8]. The intervention resulted in a small but statistically significant weight loss compared to control (−0.23 kg, (95% CI −0.89, 0.43) vs. +0.72 kg (95% CI 0.09, 1.35); group-by-treatment interaction *p* = 0.04). No other intervention effect was found on the other outcomes. This secondary analysis included participants from the intervention arm who completed the study (*n* = 206). Baseline characteristics by weight change categories are shown in Table 1. There were no statistically significant differences in the baseline characteristics between the groups. The most prevalent age categories across all groups were 30- to 34-year-olds and 35- to 39-year-olds. Most women reported being married and undertaking home duties when asked their marital status and work situation respectively. Most participants were overweight or obese at baseline in all groups. Attainment of a Bachelor degree was the most prevalent highest level of education reported across all groups. Most participants were within 1 year postpartum across all groups.

Changes in cardiometabolic measures over time within each group is shown in Table 2. Significant improvements in weight, waist and body mass index (BMI) were seen for the weight loss group (Table 2). They also showed significant improvements in glycemic control from HbA1c, and in total cholesterol, triglyceride and triglyceride:HDL ratio but not fasting plasma glucose or HDL. In the weight stable group which maintained weight within 2 kg, a small but significant decrease in waist was observed while weight and BMI remained unchanged. This was accompanied by a significant worsening of fasting plasma glucose although HbA1c was not significantly changed in this group. This group also had significant improvements in total cholesterol and LDL cholesterol but not in triglyceride. Within the weight gain group a significant deterioration in weight, BMI and waist was seen. A significant improvement in total cholesterol was seen for this group but there was also a significant deterioration in triglyceride and triglyceride:HDL ratio. HDL cholesterol decreased significantly over time in all groups.

Between-group comparisons showed significant differences in all anthropometric measures between weight loss, weight maintenance and weight gain groups (Table 2). The weight loss group had significantly smaller increase in fasting plasma glucose and significantly greater decrease in HbA1c compared with the weight gain group. The weight loss group had significantly greater decrease in total cholesterol and LDL cholesterol compared with the other two groups. There appeared to be a dose-response relationship between weight change and triglyceride with significant differences between weight loss, weight stability and weight gain groups. In terms of insulin resistance, the weight gain group had significant greater increase in triglyceride:HDL ratio, a sometimes used indicator of insulin resistance, compared with the other two groups. No time or group effect was seen in systolic and diastolic blood pressure. Similar results for the between-group comparisons were obtained after corrected for baseline weight.

## 4. Discussion

We explored the effect of weight change on anthropometric and cardiometabolic outcomes in postpartum women with a history of gestational diabetes at 1 year following a diabetes prevention program. We report that weight loss of more than 2 kg was associated with significant improvements in glycemic control, total and LDL cholesterol, triglyceride and triglyceride:HDL ratio. We found weight gain of more than 2 kg significantly worsened triglyceride and triglyceride:HDL ratio.

Diabetes prevention programs for women after gestational diabetes typically reports much smaller weight loss, averaging between 1 to 3 kg compared with 2.5 kg or more seen in “real world” diabetes prevention programs in the general population, or 5 kg in controlled trials [6,7,10,12]. The clinical significance of such small degees/amounts of weight loss on glycemic control and other metabolic outcomes is unclear. Previous studies in postpartum women after gestational diabetes with small weight loss reported no significant change in lipids [33] and fasting plasma glucose [34]. Similarly, the primary analysis of the current trial did not find a significant change to the intervention participants as a group [8]. When we stratified the group according to weight loss achieved, we found that weight loss as little as over 2 kg is associated with improvements in HbA1c, fasting plasma glucose, triglyceride:HDL, LDL cholesterol, total cholesterol and triglyceride compared with those who gained weight. This provides evidence that the small average weight loss seen in this group could still be beneficial in improving the metabolic risks.

Women of reproductive age are at high risk of weight gain [15]. Adulthood weight gain (from 18 years onwards) is associated with increased risk of chronic diseases [35]. There was little direct evidence demonstrating that weight gain increases metabolic risk in the postpartum period. The current study suggest that weight gain as little as 2 kg or more worsens triglyceride and triglyceride:HDL. triglyceride:HDL has been found to be a predictor of insulin resistance in certain populations including individuals of Aboriginal, Chinese, and European origin [25,26]. This is consistent with the known relationship between weight gain and increasing insulin resistance [36]. Considering that triglyceride:HDL was increased in weight gain but similar between weight loss and weight maintenance groups in the current analysis, this finding suggests that prevention of weight gain could prevent the progression of insulin resistance, and should be one of the diabetes prevention goals in postpartum women with history of gestational diabetes.

We report in the current analysis that improvements in total and LDL cholesterol were seen not only in the weight loss group but also in the weight maintenance and weight gain groups at 1 year after randomization to the lifestyle intervention. This suggests that improvements in metabolic health such as lipid outcomes could occur independent of weight loss, presumably due to improvements in diet quality resulting from the lifestyle intervention. The benefit of diet quality on reducing cancer, cardiovascular and all-cause mortality independent of weight change were also previously observed in the general population [37]. We also observed greater improvements of total and LDL cholesterol in the weight loss group (about 10% improvement) compared with the other groups, as consistent with previous evidence on the protective effect of weight loss on lipid metabolism in young women [21]. This is particularly relevant to women with histories of GDM as a recent meta-analysis found that GDM increases the risk of cardiovascular event by 2.3-fold in the first decade postpartum [38].

There was a decrease in HDL across all weight change groups without significant group differences. A recent meta-analysis (20 studies, *n* = 2016) on low-fat compared with high-fat diet on cardiometabolic outcomes in people without metabolic disturbance found that low-fat diet was associated with a decrease in HDL-cholesterol (WMD: −2.57 mg/dL (−0.07 mmol/L); 95% CI −3.85, −1.28; *p* < 0.001) [39]. As two of the five goals of MAGDA were on reducing total and saturated fat intake, this may have contributed to the observed change in HDL. We have also previously observed a reduction in HDL-cholesterol following a low-fat diet in women of reproductive age [21]. Weight loss should increase HDL-cholesterol, although the level of weight loss required for this is usually not seen in postpartum women or women of reproductive age [40]. Higher physical activity levels could also increase HDL-cholesterol [41] but the current group did not have a significant change in physical activity during the intervention [8].

We found that fasting plasma glucose worsened slightly over time in all weight change groups despite receiving lifestyle intervention, without reaching the diagnostic threshold for impaired fasting glucose. This is consistent with the increased risk of type 2 diabetes development associated with the history of gestational diabetes [42]. In particular, there was a significantly greater increase in fasting plasma glucose in the weight gain group compared with that in the weight loss group. This provides further evidence supporting the increased risk of type 2 diabetes with adulthood weight gain as previously reported in the Nurses’ Health Study [35]. A small but significant reduction in HbA1c was seen in the group with weight loss of 2 kg or more. HbA1c has been found to have lower sensitivity and higher specificity in detecting type 2 diabetes in at-risk populations compared to fasting plasma glucose [43]. The current findings of significantly increased fasting plasma glucose in weight gain group compared to weight loss group, and significantly decreased HbA1c in weight loss group compared to weight gain group, suggest that modest weight loss in the postpartum period could slow the progression of type 2 diabetes in this group. Longer term follow-up is required to determine if this would result in reduced incidence of diabetes. This is consistent with a previous study reporting that postpartum weight changes correspond to the incidence of type 2 diabetes among women with history of gestational diabetes [44]. This current finding on the benefit of small weight loss on glycemic improvement provide further support for modest weight loss as treatment goals in this group.

Women of reproductive age face unique barriers to lifestyle modification in terms of time, motivation, cost and social support, which may explain the high risk of weight gain in this group [15,45]. All postpartum women face additional lifestyle change barriers due to infant needs and shifts in priority [19]. Considering these significant barriers to lifestyle modification in the postpartum stage, lower weight loss or prevention of weight gain may be a more realistic and useful goal for this group. Achieving lower level weight loss or even weight stability actually constitutes a considerable achievement in light of the population trend of weight gain at this lifestage [15]. Our findings provide evidence that even a small amount of weight loss or stability in the postpartum period carries significant health benefit in women with history of GDM.

The strengths of this randomized trial include the length of follow-up and rigorous data collection methods. Limitations include this study being a secondary analysis, which is cross-sectional in nature, and as such we cannot derive causal relationships between the variables described. Also the delivery format of the intervention was resource intensive yet only achieved a low level of engagement from participants, which makes it unattractive to funders of health services.

## 5. Conclusions

We report that modest weight loss (more than 2 kg) following lifestyle intervention in postpartum women with a history of GDM was associated with improvements in HbA1c, insulin resistance (triglyceride:HDL), total cholesterol, LDL cholesterol and triglyceride compared with those who gained more than 2 kg. Significantly higher increases in fasting blood glucose, insulin resistance (triglyceride:HDL) and triglyceride were seen in those who gained more than 2 kg compared with those who maintained weight within 2 kg or lost more than 2 kg. These findings support the metabolic benefit of lower level weight loss or weight stability in postpartum women with history of gestational diabetes. Future research should develop less-intensive interventions for lower level of weight loss or weight stability for this group.

## Figures and Tables

**Figure 1 nutrients-11-00922-f001:**
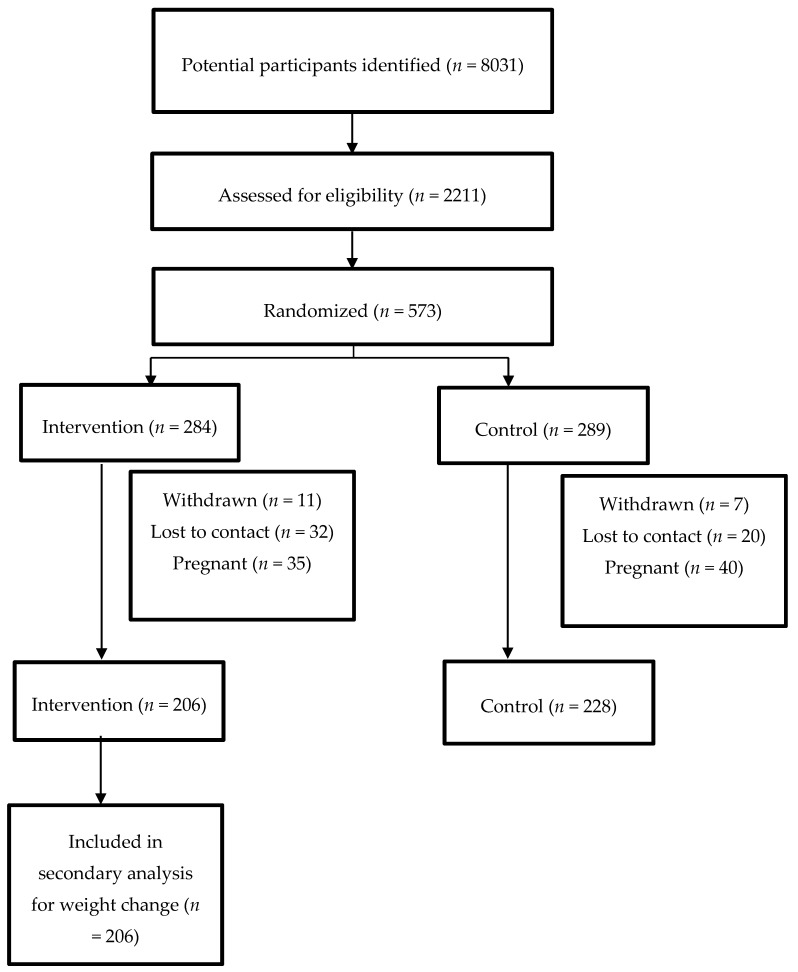
The flow of participants for the secondary analysis of the Mothers After Gestational Diabetes (MAGDA) study.

**Table 1 nutrients-11-00922-t001:** Baseline characteristics of participants.

	Weight Loss >2 kg (*n* = 74)	Weight Stable ±2 kg (*n* = 74)	Weight Gain >2 kg (*n* = 58)	*p*-Value
**Age**				
≤24	2 (2.7)	1 (1.4)	2 (3.4)	0.422
25–29	10 (13.5)	6 (8.2)	8 (13.8)	
30–34	27 (36.5)	26 (35.6)	18 (31.0)	
35–39	21 (28.4)	27 (37.0)	12 (20.7)	
≥40	14 (18.9)	13 (17.8)	18 (31.0)	
**BMI**				
<20	4 (5.6)	8 (11.0)	1 (1.7)	0.061
20 to <25	21 (29.2)	26 (35.6)	14 (24.1)	
25 to <30	22 (30.6)	13 (17.8)	12 (20.7)	
>30	25 (34.7)	26 (35.6)	31 (53.4)	
**PHQ**				
Minimal depression	43 (58.1)	50 (68.5)	40 (69.0)	0.127
Mild depression (0–9)	24 (32.4)	17 (23.3)	16 (27.6)	
Moderate depression (10–19)	7 (9.5)	3 (4.1)	2 (3.4)	
Moderately severe depression (20–27)		3 (4.1)		
**Work Situation**				
Full time	13 (17.6)	12 (16.2)	10 (17.2)	0.913
Part time	13 (17.6)	12 (16.2)	14 (24.1)	
Casual	3 (4.1)	2 (2.7)	2 (3.4)	
Home duties	34 (45.9)	36 (48.6)	22 (37.9)	
Unemployed	2 (2.7)	2 (2.7)		
Other	9 (12.2)	9 (12.2)	10 (17.2)	
**Smoking Status**				
No	72 (97.3)	72 (97.3)	54 (93.1)	0.291
Yes	2 (2.7)	1 (1.4)	4 (6.9)	
Family Income				
Low	19 (25.7)	21 (28.4)	9 (15.5)	0.124
Medium	34 (45.9)	26 (35.1)	28 (48.3)	
High	21 (28.4)	24 (32.4)	21 (36.2)	
**Highest Level Education**				
Primary	1 (1.4)	2 (2.7)		0.083
Secondary	7 (9.5)	9 (12.2)	8 (13.8)	
Certificate Level	10 (13.5)	5 (6.8)	5 (8.6)	
Diploma Level	4 (5.4)	10 (13.5)	11 (19.0)	
Bachelor Degree	37 (50.0)	30 (40.5)	22 (37.9)	
Master Degree	14 (18.9)	15 (20.3)	6 (10.3)	
Doctoral Degree	1 (1.4)		3 (5.2)	
Other		2 (2.7)	3 (5.2)	
**Time since Childbirth**				
0–26 weeks	31 (43.7)	39 (52.7)	33 (56.9)	0.578
27–52 weeks	28 (39.4)	20 (27.0)	15 (25.9)	
53–104 weeks	11 (15.5)	14 (18.9)	10 (17.2)	
>104 weeks	1 (1.4)	1 (1.4)		
**Marital Status**				
Married	62 (83.8)	53 (71.6)	47 (81.0)	0.48
Single	4 (5.4)	4 (5.4)	2 (3.4)	
Widowed				
De facto	7 (9.5)	15 (20.3)	9 (15.5)	
Divorced		1 (1.4)		
Separated	1 (1.4)			
**Breastfeeding Initiated**				
No	12 (16.2)	13 (17.6)	7 (12.1)	0.621
Yes	62 (83.8)	60 (81.1)	51 (87.9)	
Parity				
1 child	32 (43.2)	28 (38.4)	24 (41.4)	0.976
2 children	27 (36.5)	30 (41.1)	23 (39.7)	
≥3 children	15 (20.3)	15 (20.5)	11 (19.0)	
**Total Sessions Attended**				
0	19 (25.7)	19 (25.7)	14 (24.1)	0.795
1	12 (16.2)	8 (10.8)	7 (12.1)	
2	6 (8.1)	6 (8.1)	7 (12.1)	
3	5 (6.8)	6 (8.1)	2 (3.4)	
4	4 (5.4)	2 (2.7)	6 (10.3)	
5	12 (16.2)	10 (13.5)	7 (12.1)	
6	11 (14.9)	11 (14.9)	9 (15.5)	
7	4 (5.4)	7 (9.5)	5 (8.6)	
8	1 (1.4)	5 (6.8)	1 (1.7)	

BMI = body mass index; PHQ = Patient Health Questionnaire scoring. Low income = Families with one child with a before tax income of under $55,974 (plus $5224 for each extra dependent child); Medium income = Families with one child with a before tax income greater than $55,974 but less than $94,339 (plus $5224 for each extra dependent child); High income = Families with one child with a before tax income of more than $94,539 (plus $5028 for each extra dependent child). Data presented as frequency and percentage. *p* = value indicate differences in the proportions between groups were analyzed using the chi-square test.

**Table 2 nutrients-11-00922-t002:** Anthropometric and cardiometabolic outcomes by weight change at 1-year during the Mothers After Gestational Diabetes in Australia intervention.

	Weight Loss >2 kg (*n* = 74)				Weight Stability ±2 kg (*n* = 74)				Weight Gain >2 kg (*n* = 58)					
	Baseline	12 Months	Change	*p*-Value ^1^	Baseline	12 Months	Change	*p*-Value ^1^	Baseline	12 Months	Change	*p*-Value ^1^	*p*-Value ^2^	*p*-Value ^3^
BMI, kg/m^2^	28.4 ± 5.8	26.5 ± 5.6	−1.9 ± 1.1 ^a^	<0.001	28.0 ± 7.5	28.1 ± 7.5	0.1 ± 0.5 ^b^	0.191	31.5 ± 7.8	33.4 ± 8.2	1.9 ± 1.0 ^c^	<0.001	<0.001	<0.001
Weight, kg	73.5 ± 18.6	68.7 ± 17.9	−4.9 ± 2.9 ^a^	<0.001	73.5 ± 20.8	73.7 ± 20.8	0.2 ± 1.2 ^b^	0.153	83.5 ± 21.4	88.7 ± 23.0	5.2 ± 3.0 ^c^	<0.001	<0.001	<0.001
Waist, cm	89.2 ± 12.1	83.7 ± 12.1	−5.4 ± 5.1 ^a^	<0.001	90.6 ± 15.2	88.8 ± 15.5	−1.8 ± 4.6 ^b^	0.001	96.5 ± 15.6	97.9 ± 16.1	1.4 ± 4.6 ^c^	0.026	<0.001	<0.001
FPG, mmol/L	4.8 ± 0.5	4.9 ± 0.5	0.1 ± 0.4 ^a^	0.074	4.8 ± 0.6	5.0 ± 0.7	0.2 ± 0.5 ^ab^	0.001	4.8 ± 0.5	5.1 ± 0.7	0.3 ± 0.6 ^bc^	0.001	0.101	0.076
HbA1c, %	5.30 ± 0.4	5.18 ± 0.4	−0.12 ± 0.4 ^a^	0.004	5.40 ± 0.4	5.34 ± 0.5	−0.06 ± 0.5 ^ab^	0.292	5.33 ± 0.4	5.35 ± 0.4	0.02 ± 0.4 ^bc^	0.658	0.144	0.141
Tchol, mmol/L	5.2 ± 0.9	4.7 ± 0.9	−0.6 ± 0.8 ^a^	<0.001	5.1 ± 0.9	4.8 ± 0.8	−0.2 ± 0.7 ^b^	0.007	5.2 ± 1.0	4.9 ± 0.8	−0.2 ± 0.6 ^b^	0.005	0.005	0.004
LDL, mmol/L	3.2 ± 0.9	2.8 ± 0.8	−0.4 ± 0.6 ^a^	<0.001	3.1 ± 0.8	2.9 ± 0.7	−0.2 ± 0.6 ^b^	0.044	3.0 ± 0.9	2.9 ± 0.8	−0.1 ± 0.7 ^b^	0.238	0.017	0.014
HDL, mmol/L	1.5 ± 0.3	1.4 ± 0.3	−0.1 ± 0.3 ^a^	0.014	1.4 ± 0.4	1.3 ± 0.4	−0.1 ± 0.2 ^a^	0.014	1.5 ± 0.3	1.3 ± 0.3	−0.2 ± 0.3 ^a^	<0.001	0.884	0.891
TG, mmol/L	1.3 ± 0.7	1.1 ± 0.5	−0.2 ± 0.5 ^a^	0.001	1.3 ± 0.8	1.3 ± 0.6	0 ± 0.6 ^b^	0.984	1.2 ± 0.6	1.4 ± 0.7	0.3 ± 0.5 ^c^	<0.001	<0.001	<0.001
TG/HDL	1.0 ± 0.7	0.9 ± 0.6	−0.1 ± 0.4 ^a^	0.017	1.1 ± 0.9	1.1 ± 0.8	0 ± 0.6 ^a^	0.625	0.9 ± 0.7	1.2 ± 0.8	0.4 ± 0.7 ^b^	<0.001	<0.001	<0.001
SBP, mmHg	111.4 ± 11.5	112.0 ± 10.6	0.7 ± 9.5 ^a^	0.536	113.1 ± 14.4	112.0 ± 13.2	−1.1 ± 9.5 ^a^	0.338	113.7 ± 12.8	114.9 ± 10.5	1.2 ± 12.3 ^a^	0.460	0.407	0.298
DBP, mmHg	70.4 ± 9.3	72.3 ± 9.5	2 ± 8.5 ^a^	0.052	71.1 ± 10.6	71.8 ± 11.3	0.7 ± 7.3 ^a^	0.422	72.0 ± 8.6	73.4 ± 8.4	1.5 ± 7.9 ^a^	0.161	0.614	0.620

FPG = fasting plasma glucose; HbA1c = glycated haemoglobin; Tchol = total cholesterol; LDL = low-density lipoprotein cholesterol; HDL = high-density lipoprotein cholesterol; TG = triglyceride; TG/HDL = triglyceride-to-high-density-lipoprotein-cholesterol ratio; SBP = systolic blood pressure; DBP = diastolic blood pressure. Data presented as mean + SD. Change over time within group was assessed using paired *t*-test (*p*-value ^1^). Changes between the groups were assessed using analysis of variance (*p*-value ^2^). Changes between the groups were assessed using analysis of covariance with baseline weight as a covariate (*p*-value ^3^). Differences between the groups were identified using superscripts (^a,b,c^). Common superscript letters within a row denote that means were not different (*p* < 0.05).

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
