# Peer review of "Weight Change and Cardiometabolic Outcomes in Postpartum Women with History of Gestational Diabetes"

_nutrients, 2019, doi:10.3390/nu11040922_

Round 1

Reviewer 1 Report

This manuscript is a nice use of secondary data to provide a greater understanding of the impacts of a behavioral intervention study. Overall the manuscript is well-written and flows logically. My specific comments about areas of the manuscript I believe may be improved are as follows:

Methods:

The description of the MAGDA Trial in the Methods section was generally very thorough with the exception of the description of the measurements (Section 2.5, P3L137). The measurement section would benefit from more detail as the measurements are the primary outcome and clarification about the procedures is helpful to interpret the results.

Discussion/Conclusions:

While weight loss or weight maintenance showed benefits for many of the cardiometabolic markers, the finding that FPG did not significantly decline in any group and significantly increased in the weight stability and weight gain groups merits further discussion. It would be particularly helpful to discuss the significant beneficial impacts on HbA1c in contrast and expand on possible reasons why the glycemic results differ.

Author Response

Response to Reviewer 1 Comments

 Point 1: The description of the MAGDA Trial in the Methods section was generally very thorough with the exception of the description of the measurements (Section 2.5, P3L137). The measurement section would benefit from more detail as the measurements are the primary outcome and clarification about the procedures is helpful to interpret the results.

 Response 1: This section has now been expanded as below with an additional reference to the protocol for clinical measurements (p4, ln 138-149):

‘The protocol for measurements was described in detail in previous publication [1]. In brief, participants completed a survey that include demographic questions at baseline. Anthropometric and blood pressure measurements were completed by trained study nurses to ensure that measurements meet research standards. Blood samples were collected by study nurse or phlebotomists and analyzed by private pathology centers including Melbourne Pathology (Victoria) or Clinpath Laboratories (South Australia). All clinical measures including height, weight, waist circumference and blood pressure were measured using according to the recommendations of the European Health Risk Monitoring protocol [2]. Fasting venous blood samples were collected after an overnight fast from 10 pm the night before. Blood samples were analyzed for triglycerides, total cholesterol, low-density lipoprotein cholesterol and high-density lipoprotein cholesterol, HbA1c, and fasting glucose and 2-hour glucose tolerance as previously described [1,3]. ‘

 Point 2: While weight loss or weight maintenance showed benefits for many of the cardiometabolic markers, the finding that FPG did not significantly decline in any group and significantly increased in the weight stability and weight gain groups merits further discussion.

 Response 2: This section has been expanded as below to discuss this important finding (pg 11, ln 286-292):

‘We found that fasting plasma glucose worsened slightly over time in all weight change groups despite receiving lifestyle intervention, without reaching the diagnostic threshold for impaired fasting glucose. This is consistent with the increased risk of type 2 diabetes development associated with the history of gestational diabetes [4]. In particular, there was a significantly greater increase in fasting plasma glucose in the weight gain group compared with that in the weight loss group. This provides further evidence supporting the increased risk of type 2 diabetes with adulthood weight gain as previously reported in the Nurses’ Health Study [5].’

 Point 3: It would be particularly helpful to discuss the significant beneficial impacts on HbA1c in contrast and expand on possible reasons why the glycemic results differ.

 Response 3: The following has been added to discuss this finding (pg 11, ln 293-298):

‘A small but significant reduction in HbA1c was seen in the group with weight loss of 2kg or more. HbA1c has been found to have lower sensitivity and higher specificity in detecting type 2 diabetes in at-risk populations compared to fasting plasma glucose [6]. The current findings of significantly increased fasting plasma glucose in weight gain group compared to weight loss group, and significantly decreased HbA1c in weight loss group compared to weight gain group suggest that modest weight loss in the postpartum period could slow the progression of type 2 diabetes in this group.’

References

1.            Shih, S.T.F.; Davis-Lameloise, N.; Janus, E.D.; Wildey, C.; Versace, V.L.; Hagger, V.; Asproloupos, D.; O'Reilly, S.; Phillips, P.A.; Ackland, M., et al. Mothers after gestational diabetes in australia diabetes prevention program (magda-dpp) post-natal intervention: Study protocol for a randomized controlled trial. Trials 2013, 14.

2.            Tolonen, H.; Kuulasmaa, K.; Laatikainen, T.; Wolf, H.; Project, E.H.R.M. Recommendations for indicators, international collaboration, protocol and manual operations chronic disease risk factor surveys; Finnish National Public Health Institute: Helsinki, 2002.

3.            O'Reilly, S.L.; Dunbar, J.A.; Versace, V.; Janus, E.; Best, J.D.; Carter, R.; Oats, J.J.; Skinner, T.; Ackland, M.; Phillips, P.A., et al. Mothers after gestational diabetes in australia (magda): A randomised controlled trial of a postnatal diabetes prevention program. PLoS Med 2016, 13, e1002092.

4.            Kim, C.; Newton, K.M.; Knopp, R.H. Gestational diabetes and the incidence of type 2 diabetes - a systematic review. Diabetes Care 2002, 25, 1862-1868.

5.            Zheng, Y.; Manson, J.E.; Yuan, C.; Liang, M.H.; Grodstein, F.; Stampfer, M.J.; Willett, W.C.; Hu, F.B. Associations of weight gain from early to middle adulthood with major health outcomes later in life. JAMA 2017, 318, 255-269.

6.            Valderhaug, T.G.; Sharma, A.; Kravdal, G.; Ronningen, R.; Nermoen, I. The usage of fasting glucose and glycated hemoglobin for the identification of unknown type 2 diabetes in high risk patients with morbid obesity. Scand J Clin Lab Invest 2017, 77, 505-512.

7.            Melzer, K.; Schutz, Y. Pre-pregnancy and pregnancy predictors of obesity. Int J Obes (Lond) 2010, 34 Suppl 2, S44-52.

8.            Nicklas, J.M.; Zera, C.A.; Seely, E.W. Predictors of very early postpartum weight loss in women with recent gestational diabetes mellitus. J Matern Fetal Neonatal Med 2018, 1-7.

9.            Australian Institute of Health and Welfare, A. 2010 australian national infant feeding survey: Indicator results; 2010.

10.          Lu, M.Q.; Wan, Y.; Yang, B.; Huggins, C.E.; Li, D. Effects of low-fat compared with high-fat diet on cardiometabolic indicators in people with overweight and obesity without overt metabolic disturbance: A systematic review and meta-analysis of randomised controlled trials. Brit J Nutr 2018, 119, 96-108.

11.          Lim, S.S.; Norman, R.J.; Clifton, P.M.; Noakes, M. The effect of comprehensive lifestyle intervention or metformin on obesity in young women. Nutr Metab Cardiovasc Dis 2011, 21, 261-268.

12.          Dansinger, M.; Williams, P.T.; Superko, H.R.; Asztalos, B.F.; Schaefer, E.J. Effects of weight change on hdl-cholesterol and its subfractions in over 28,000 men and women. J Clin Lipidol 2018.

13.          Zwald, M.L.; Akinbami, L.J.; Fakhouri, T.H.; Fryar, C.D. Prevalence of low high-density lipoprotein cholesterol among adults, by physical activity: United states, 2011-2014. NCHS Data Brief 2017, 1-8.

Reviewer 2 Report

In the manuscript, "Weight change and cardiometabolic outcomes in postpartum women with history of gestational diabetes" Lim and colleagues have undertaken a secondary analysis to investigate cardiometabolic outcomes in with with GDM undergoing lifestyle interventions post pregnancy. They have shown that the lipid profiles are improved in those that lost weight or did not gain weight. This is novel finding with significant importance, but a few concerns are raised.

The groups were divided into those that lost weight, had stable weight, or gained weight. In looking at the average weight in the groups. The group that gained weight appears to have increased weight at baseline. A statistical analysis should be included for this measure and it if is different, it should be controlled for in future analysis. 

In the cohort, many of the women are breastfeeding. This will alter metabolism. This should also be controlled in models of the results. The duration of breastfeeding should also be included in available. The large percentage of participants that report breastfeeding is unusual as many mothers with GDM are not successful with lactation.

The authors point out that acceptance of the intervention as not well received (25% in each group never attended a session). It would be interesting to know which sessions were attended and if a particular session was associated with the alterations in cholesterol.

It is noted that in all groups there was a decrease in HDL. Was any assessment made of physical activity? The decline in HDL should also receive a comment in the discussion since it was a significant. 

The authors report the change in A1c was significant in the weight loss group but not in the weight stable group. The change and SD are very similar between the 2 groups. If A1c was not normally distributed a nonparametric test would be needed to assess difference. The authors should review this statistical analysis.

In Table 2. superscripts of a and b are used in the table, but no clear description of their meaning is given. a and b needed to be defined in the table legend or removed.

An extra space is noted in line 24.

Author Response

Response to Reviewer 2 Comments

 Point 1: The groups were divided into those that lost weight, had stable weight, or gained weight. In looking at the average weight in the groups. The group that gained weight appears to have increased weight at baseline. A statistical analysis should be included for this measure and it if is different, it should be controlled for in future analysis.

In the cohort, many of the women are breastfeeding. This will alter metabolism. This should also be controlled in models of the results. The duration of breastfeeding should also be included in available.

 Response 1: We reported in Table 1 that baseline BMI and breastfeeding were not statistically significant between the groups.

We acknowledge that there are a number of factors that could potentially explain the mechanisms underlying the observed relationships between weight change and metabolic outcomes reported in this study. To further this work, we intend to look at predictive models that include baseline weight, weight change, diet, physical activity, ethnicity, socioeconomic status, depression, breastfeeding duration, smoking, age, and parity for on metabolic outcomes [7,8]. These were identified as potential confounders in postpartum weight loss although there were no consistently significant predictors in every study population [7,8]. We intend to pursue this research question in subsequent work as it is beyond the scope of the current paper to address all these in full.

 Point 2: The large percentage of participants that report breastfeeding is unusual as many mothers with GDM are not successful with lactation.

 Response 2: This measure indicated any initiated breastfeeding, not continued nor exclusive breastfeeding. The label has been changed to ‘Breastfeeding initiated’ for clarity (Table 1).

The rate of initiating breastfeeding in the current study is lower than the general population of Australia [9], in agreement with the observation by the reviewer.

 Point 3: The authors point out that acceptance of the intervention as not well received (25% in each group never attended a session). It would be interesting to know which sessions were attended and if a particular session was associated with the alterations in cholesterol.

 Response 3: This is indeed an interesting exploration. We could add intervention session to the list of predictors in the future analysis mentioned above. That is a different research question from the current study which seeks to describe the metabolic consequences of weight maintenance vs modest weight loss vs weight gain so that we could provide evidence-based clinical advice on modest weight loss and weight maintenance to postGDM mothers.

 Point 4: It is noted that in all groups there was a decrease in HDL. Was any assessment made of physical activity? The decline in HDL should also receive a comment in the discussion since it was a significant.

 Response 4: The following has been added to discuss this significant finding:

‘There was a decrease in HDL across all weight change groups without significant group differences. A recent meta-analysis (20 studies, n=2016) on low-fat compared with high-fat diet on cardiometabolic outcomes in people without metabolic disturbance found that low-fat diet was associated with a decrease in HDL-cholesterol (WMD: -2·57 mg/dl or -0·07 mmol/l); 95 % CI -3·85, -1·28; P<0·001 [10]. As two of the five goals of MAGDA were on reducing total and saturated fat intake, this may have contributed to the observed change in HDL. We have also previously observed a reduction in HDL-cholesterol following a low-fat diet in women of reproductive age [11]. Weight loss should increase HDL-cholesterol, although the level of weight loss required for this is usually not seen in postpartum women or women of reproductive age [12]. Higher physical activity levels could also increase HDL-cholesterol [13] but the current group did not have a significant change in physical activity during the intervention [3].

 Point 5: The authors report the change in A1c was significant in the weight loss group but not in the weight stable group. The change and SD are very similar between the 2 groups. If A1c was not normally distributed a nonparametric test would be needed to assess difference. The authors should review this statistical analysis.

 Response 5: This was checked by our biostatistician and the apparent similarity in change was due to rounding down to one decimal place for consistency in formatting for the table. The change in the weight loss group was -0.123 (0.4, p=0.004) and the change in the weight stable group was -0.063 (0.5, p=0.292). The results for HbA1c have now been revised to two decimal places (Table 2).

 Point 6: In Table 2. superscripts of a and b are used in the table, but no clear description of their meaning is given. a and b needed to be defined in the table legend or removed.

An extra space is noted in line 24.

 Response 6: The superscripts were used to identify differences between groups in post-hoc test. We have now clarified this in the footnotes as below:

‘Differences between the groups were identified using superscripts (a, b, c). Change values with different superscripts are significantly different from each other (P<0.05).’

References

1.            Shih, S.T.F.; Davis-Lameloise, N.; Janus, E.D.; Wildey, C.; Versace, V.L.; Hagger, V.; Asproloupos, D.; O'Reilly, S.; Phillips, P.A.; Ackland, M., et al. Mothers after gestational diabetes in australia diabetes prevention program (magda-dpp) post-natal intervention: Study protocol for a randomized controlled trial. Trials 2013, 14.

2.            Tolonen, H.; Kuulasmaa, K.; Laatikainen, T.; Wolf, H.; Project, E.H.R.M. Recommendations for indicators, international collaboration, protocol and manual operations chronic disease risk factor surveys; Finnish National Public Health Institute: Helsinki, 2002.

3.            O'Reilly, S.L.; Dunbar, J.A.; Versace, V.; Janus, E.; Best, J.D.; Carter, R.; Oats, J.J.; Skinner, T.; Ackland, M.; Phillips, P.A., et al. Mothers after gestational diabetes in australia (magda): A randomised controlled trial of a postnatal diabetes prevention program. PLoS Med 2016, 13, e1002092.

4.            Kim, C.; Newton, K.M.; Knopp, R.H. Gestational diabetes and the incidence of type 2 diabetes - a systematic review. Diabetes Care 2002, 25, 1862-1868.

5.            Zheng, Y.; Manson, J.E.; Yuan, C.; Liang, M.H.; Grodstein, F.; Stampfer, M.J.; Willett, W.C.; Hu, F.B. Associations of weight gain from early to middle adulthood with major health outcomes later in life. JAMA 2017, 318, 255-269.

6.            Valderhaug, T.G.; Sharma, A.; Kravdal, G.; Ronningen, R.; Nermoen, I. The usage of fasting glucose and glycated hemoglobin for the identification of unknown type 2 diabetes in high risk patients with morbid obesity. Scand J Clin Lab Invest 2017, 77, 505-512.

7.            Melzer, K.; Schutz, Y. Pre-pregnancy and pregnancy predictors of obesity. Int J Obes (Lond) 2010, 34 Suppl 2, S44-52.

8.            Nicklas, J.M.; Zera, C.A.; Seely, E.W. Predictors of very early postpartum weight loss in women with recent gestational diabetes mellitus. J Matern Fetal Neonatal Med 2018, 1-7.

9.            Australian Institute of Health and Welfare, A. 2010 australian national infant feeding survey: Indicator results; 2010.

10.          Lu, M.Q.; Wan, Y.; Yang, B.; Huggins, C.E.; Li, D. Effects of low-fat compared with high-fat diet on cardiometabolic indicators in people with overweight and obesity without overt metabolic disturbance: A systematic review and meta-analysis of randomised controlled trials. Brit J Nutr 2018, 119, 96-108.

11.          Lim, S.S.; Norman, R.J.; Clifton, P.M.; Noakes, M. The effect of comprehensive lifestyle intervention or metformin on obesity in young women. Nutr Metab Cardiovasc Dis 2011, 21, 261-268.

12.          Dansinger, M.; Williams, P.T.; Superko, H.R.; Asztalos, B.F.; Schaefer, E.J. Effects of weight change on hdl-cholesterol and its subfractions in over 28,000 men and women. J Clin Lipidol 2018.

13.          Zwald, M.L.; Akinbami, L.J.; Fakhouri, T.H.; Fryar, C.D. Prevalence of low high-density lipoprotein cholesterol among adults, by physical activity: United states, 2011-2014. NCHS Data Brief 2017, 1-8.

Round 2

Reviewer 2 Report

Lim and colleagues have reported in their manuscript the effect of weight loss or weight maintenance on cardiovascular outcomes in women post-pregnancy. While some of the concerns are addressed, a few elements still require attention. 

The authors mention that BMI is not different; however, weight is different and needs to be evaluated in a multiple regression model for the impact on cardiometabolic measures.

The authors have included superscripts in the legend for Table 2, but have not defined what they denote. This must be included.

The authors also relate the changes in lipids to changes in diet. If this could be associated with the women who specifically attended a session educating about dietary fat, this might bolster this argument.

Author Response

Thank you for the comments. Please see the response below to address the comments.

Comments

Response

The   authors mention that BMI is not different; however, weight is different and   needs to be evaluated in a multiple regression model for the impact on cardiometabolic   measures.

We   have added ANCOVA analyses to correct for baseline weight.

This   has been added to the Methods section (p4 ln 162-163):  

The differences in   changes between the three groups were analyzed using one-way ANCOVA with   baseline weight as a covariate.’

This has   been added to the Results section (p8, ln 233-234):

‘Similar   results for the between-group comparisons were obtained after corrected for   baseline weight.’

The   authors have included superscripts in the legend for Table 2, but have not   defined what they denote. This must be included.

We   have now rephrased the legend to below for greater clarity:

‘Common superscript letters   within a row denote that means were not different   (P<0.05)’

The   authors also relate the changes in lipids to changes in diet. If this could   be associated with the women who specifically attended a session educating   about dietary fat, this might bolster this argument.

Our   biostatistician ran the analysis on session attendance on weight outcomes and   found no statistical significance. He has yet to conduct the analysis on   lipid outcomes. We are also not intending to explore predictors on other   outcomes (e.g. session on glycemic index and glycemic outcomes). Further,   there is overlap in content across the sessions, e.g. Session 3 and 4 provide   skills to apply the content from Session 1 and 2, so it is unclear if the   proposed analysis would be entirely meaningful.

If   the reviewer insists on this analysis (Session 1 and 2 on lipid outcomes),   could we please receive an extension to provide the response? Our   biostatistician is currently on leave until the 29th April, from   which I will be on leave until the 13th May. To include this   analysis, could we please have an extension to 16th May?